# Prevalence and Genotype-Phenotype Correlation of Lynch Syndrome in a Selected High-Risk Cohort from Qatar’s Population

**DOI:** 10.3390/genes13112176

**Published:** 2022-11-21

**Authors:** Mariem Sidenna, Houssein Khodjet-El-khil, Hajar Al Mulla, Mashael Al-Shafai, Hind Hassan Habish, Reem AL-Sulaiman, Salha Bujassoum Al-Bader

**Affiliations:** 1Department of Adult and Pediatric Medical Genetics, Hamad Medical Corporation, Doha 3050, Qatar; 2Department of Cancer Genetics, Medical Oncology, National Center of Cancer Care and Research, Hamad Medical Corporation, Doha 3050, Qatar; 3Department of Biomedical Science, College of Health Sciences, QU Health, Qatar University, Doha 2713, Qatar

**Keywords:** Lynch syndrome, mismatch repair genes, colorectal cancer, pathogenic variants, Qatar

## Abstract

Lynch syndrome (LS) is the most common cause of hereditary colorectal cancers (CRC) and is associated with an increased risk for ovarian and endometrial cancers. There is lack of knowledge on the epidemiology of LS in the non-Caucasian populations especially in Qatar. The aim of this retrospective study is to explore the prevalence of LS in a selected high-risk cohort in the State of Qatar in addition to investigating the frequency and genotype-phenotype correlation associated with mismatch repair genes pathogenic variants. Retrospective review of medical records of 31 individuals with LS, 20 affected with colorectal cancer and 11 unaffected with family history of cancers, referred from January 2017 until August 2020. The prevalence of LS among affected and unaffected patients is 22% (20/92) and 2.2% respectively. Among affected individuals, *MLH1* and *MSH2* genes were highly frequent while for unaffected individuals, a recurrent *PMS2* pathogenic variant was reported in several related individuals suggesting a tribal effect. This study highlights the epidemiology of LS in high-risk cohort in Qatar which helps to provide recommendations on genetic testing, and personalize surveillance and management programs

## 1. Introduction

Lynch syndrome (LS) is the main cause of inherited colorectal cancer, it also accounts for an elevated risk of other extracolonic cancers such as gastric, endometrial and ovarian cancers [1]. Hereditary Non-Polyposis Colorectal Cancer (HNPCC) was the former name of Lynch syndrome which is associated with 1–5% of all colorectal cancer cases [2], about 3% of endometrial cancers (EC) [3], and 2% of ovarian cancers (OC) [4].

LS is inherited in an autosomal dominant manner and results from pathogenic variants in the DNA mismatch repair genes (MMR genes), (*MLH1, MSH2, MSH6*, and *PMS2*) [5]. Furthermore, germline deletions in *EPCAM* gene have been linked to Lynch syndrome [6]. *EPCAM* gene does not belong to the MMR genes, yet, pathogenic deletions can disturb its 3′ end which results in epigenetic inactivation of its neighboring gene (*MSH2*) that would be silenced [6].

Individuals who are suspected to have Lynch syndrome are often assessed for meeting revised Amsterdam II [5] or revised Bethesda criteria [7]. These criteria focus on the presence of a strong family history of Lynch syndrome associated cancers, young age at diagnosis and specific histopathological characteristics [8]. However, they have been associated with a limited sensitivity and specificity [9]. Additionally, Microsatellite instability (MSI) and Immunohistochemistry (IHC) tests are used to assess for the eligibility of genetic testing for LS. Defective MMR genes cause a variation in the microsatellites repeat number in the tumor tissue as compared to normal tissues in the same individual which can be detected through MSI testing [10]. Immunohistochemistry (IHC) is a tool to detect the loss of MMR protein expression for colon and endometrial cancers [10]. However, the interaction between MMR proteins could affect the sensitivity of immunohistochemistry in the detection of individual proteins. For instance, *MSH2* creates heterodimers with *MSH6* [11,12], thus negative staining for both proteins is common in *MSH2* pathogenic variant carriers [13]. Likewise, for *MLH1* pathogenic variant carriers, IHC often shows an absence of both MLH1 and PMS2 proteins since the two form heterodimers [14].

The results of the international Mismatch Repair Consortium (IMRC) show that *MLH1* and *MSH2* genes are responsible for 76.5% of all of the mutations identified while *MSH6* pathogenic variants account for 15.12% of cases followed by *PMS2* and *EPCAM* genes which accounts for 7.7% and 0.65 respectively of the identified cases [15].

Several studies have investigated genotype-phenotype correlation of MMR genes in Lynch Syndrome [7,16,17,18]. For instance, *MSH2* pathogenic variants carriers presented with a higher risk of metachronous and synchronous colon tumors as compared to *MLH1* pathogenic variants carriers in according to a Chinese study [19]. Also, a study conducted on one of the largest HNPCC cohorts in Germany found that *MSH2* pathogenic carriers had a late-onset endometrial cancer besides a lower percentage of CRC in *MLH1* female carriers as compared to male carriers [16].

Nevertheless, the data gained from these studies have been restricted to Caucasian families coming from European descent resulting in a gap in the comprehension of genetic epidemiology and genotype-phenotype correlation in Lynch Syndrome among non-Caucasian families.

The State of Qatar is a Middle Eastern country situated on the northeastern shore of the Arabian Peninsula, encircled by the Arabian/Persian Gulf. The number of inhabitants in Qatar frames a multiethnic local area; around 600,000 (22%) of the 2.7 million population are local Qataris (Ministry of Development Planning and Statistics); expatriates make up the excess part of the population [20]. Lynch Syndrome is the second most common cancer syndrome in Qatar after Hereditary breast and ovarian cancer (HBOC) [21]. There is a lack of studies on the epidemiology and genotype-phenotype correlation in Lynch syndrome in Qatar. While most of Lynch syndrome studies in the literature are population-based studies, our approach is more clinically oriented focusing on patients affected with colorectal cancer or unaffected individuals with family history indicative for Lynch syndrome The aim of our study is to explore the prevalence of Lynch syndrome, frequency of MMR genes pathogenic variants and the genotype-phenotype correlation of Lynch syndrome in colorectal cancer in a selected high risk cohort of ethnically diverse individuals in Qatar.

## 2. Materials and Methods

### 2.1. Study Design

The study was conducted by retrospectively reviewing the medical and genetic records of all patients referred to high risk genetic clinic at National Center for Cancer Care and Research hospital (if they meet Amsterdam/Bethesda guidelines, have an extensive family history of cancers or have a known familial mutation) from the period of January 2017 until August 2020.

A total number of 592 individuals were filtered according to our inclusion/exclusion criteria, all of them were tested for MMR genes, including 92 patients affected by colon cancer and have high risk features of LS and 500 unaffected individuals with positive family history of colon cancer and indicative of Lynch syndrome.

Review of data revealed 31 individuals including 20 patients found to be positive for Lynch syndrome among the 92 high-risk patients tested for MMR genes and 11 unaffected individuals positive for lynch syndrome among the 500 unaffected individuals who opted for genetic testing. The remaining 561 individuals were negative for lynch syndrome and were excluded as they did not have pathogenic variants in MMR genes (Figure 1).

### 2.2. Study Participants

All individuals with MMR/*EPCAM* genes pathogenic/ likely pathogenic variants were included in the study and were divided into 2 categories affected and unaffected:

Affected individuals are defined as patients with colorectal cancer and harboring pathogenic MMR gene variants who were referred due to meeting Amsterdam/Bethesda criteria.

Unaffected individuals are defined as individuals not affected with cancer and harboring pathogenic MMR genes variants referred due to meeting one or more of Bethesda guidelines, family history of cancers with young ages at onset, and/or a known familial variant. (Figure 1).

### 2.3. Medical Records and Genetic Testing Reports

Epidemiological information in addition to clinical information including patient’s age, gender, ethnic background, personal cancer history (Including primary site, age at diagnosis, histopathology, immunohistochemistry, grade, stage, lymph node involvement have been collected from the participants medical records.

Genetic tests (using either Comprehensive Cancer Panel (48 genes), Lynch/Colorectal High-risk Panel (7 genes), Colorectal Cancer Panel (21 genes), Breast/Gyni Cancer Panel (24 genes), Common Cancer Management Panel (39 genes) or Targeted Variant Testing) were done at GeneDX lab (https://www.genedx.com/, accessed on 15 May 2021). Genetic testing results were collected from the genetic reports of each individual. These include variants in MMR genes (*MLH1, MSH2, MSH6*, and *PMS2*) and *EPCAM* gene and were used to calculate the frequency of pathogenic variants in these genes among individuals with Lynch Syndrome (affected and unaffected). It was also used to compare their frequencies between affected and unaffected individuals.

The novelty of the variants was reported in the individuals’ genetic test reports and was also assessed using Clinvar (https://www.ncbi.nlm.nih.gov/clinvar/, accessed on 21 February 2021), HGMD (http://www.hgmd.cf.ac.uk/ac/index.php, accessed on 21 February 2021) and Gene Cards (https://www.genecards.org/, accessed on 21 February 2021).

### 2.4. Prevalence Calculation

Prevalence of Lynch Syndrome in affected individuals was Calculated by dividing the total number of affected individuals (individuals diagnosed with colorectal cancer and harboring MMR pathogenic variants) by the total number of high-risk affected individuals with and without the MMR pathogenic variants.

Prevalence of Lynch Syndrome in unaffected individuals was calculated by dividing the total number of unaffected individuals harboring MMR pathogenic variants by the total number of high-risk unaffected individuals with and without MMR pathogenic variants.

### 2.5. Genotype-Phenotype Correlation and Statistical Analysis

Genotype-phenotype correlation between the clinicopathological data with the type of mutated gene and with the type of variant were investigated. Correlation between Immunohistochemical staining of MMR proteins versus germline genetic test results was performed. All correlations have been assessed using Fisher’s exact test. They were performed using IBM SPSS statistical software (version 19) for data management. All categorical and binary variables were analyzed by Fischer’s exact test (*p* = ((a + b) ! (c + d) ! (a + c) ! (b + d) ! )/a ! b ! c ! d ! N !).

A two-sided *p* value < 0.05 was considered statistically significant. A post hoc test has been performed for significant correlations observed by Fisher’s exact test. Bonferroni correction was used for statistically significant post hoc test values. For the distribution of MMR and *EPCAM* genes variants in affected and unaffected individuals, a *p*-value < 0.005 (0.05/10) was considered statistically significant, while for the correlation of CRC side vs the type of variant a *p*-value <0.006 (0.05/8) was considered to be significant.

### 2.6. Variants Classification

Pathogenic/ Likely pathogenic variants in MMR*/EPCAM* genes (in affected and unaffected individuals) have been classified by definition into point mutations (nonsense and missense mutations), Large deletions, Frameshift mutations, and Intronic mutations.

## 3. Results

The study includes a total of 31 individuals harboring pathogenic variants in MMR genes (20 affected and 11 unaffected). (Figure 1).

Of the 20 affected patients, 14 were males (70%) and 6 were females (30%), with an average age of 48.2 ± 11.2 years. Colon cancer was predominantly left-sided (in 12 out of the 20 patients; 60%). Most of the cancers were in stage 2 (60%) and 14 out of the 20 patients (70%) were diagnosed before the age of 50 years. Positive cancer family history was seen in 15 patients (75%) (Table 1). Regarding the 11 unaffected high-risk individuals, 8 were females (72%) and 3 were males (27.27%) with an average age of 45.2 ± 13.5 years.

The prevalence of Lynch syndrome was calculated in a selected high-risk population of colon cancer patients (selected based on meeting one or more Amsterdam and Bethesda criteria) and in high-risk unaffected individuals (based on meeting one or more Bethesda guidelines, having a family history of cancers and/or a known pathogenic variant in MMR gene in the family). In affected patients, the prevalence of LS was 22% (20/92), while in unaffected individuals it was 2.2%.

### 3.1. Frequency of MMR Pathogenic Variants

Genetic data was collected for variants in MMR genes (*MLH1, MSH2, MSH6*, and *PMS2*) as well as for germline deletions in *EPCAM* gene. The distribution of MMR genes and *EPCAM* gene variants was significantly different between LS affected and unaffected individuals (*p*-value < 0.001 ^a^) (Table 2).

In LS affected individuals, *MLH1* and *MSH2* were the most commonly reported genes accounting for almost 90% of the variants and they were exclusively reported in affected individuals (*p*-value = 0.009) which was not statistically significant after Bonferroni correction while *MSH6* and *PMS2* genes were the least the least reported with 5% each.

For unaffected LS individuals however, the most reported gene was *PMS2* accounting for almost 63.6% of the variants (*p*-value = 0.0001) which is statistically significant after Bonferroni correction, followed by *MSH6* and *EPCAM* genes with each accounting for 18.2% of the variants (Table 2).

### 3.2. Types of Variants in MMR and EPCAM Genes

In LS affected individuals, point variants and frameshift variants were the most commonly seen variants in *MLH1* and *MSH2* genes accounting for 66.7% of each variant. For *MSH6* gene, one frameshift was reported and for *PMS2* gene, only one point mutation was reported (Table 3).

In LS unaffected individuals, most of the pathogenic variants in *PMS2* were large deletions (71.4%) followed by point mutations (14.3%) and intronic variants (14.3%) (Table 4). A heterozygous deletion encompassing exons 6 to 11 in *PMS2* gene was the most commonly reported pathogenic variant in unaffected individuals and all four carriers (heterozygous) of this pathogenic variant belong to the same tribe. For *MSH6* gene, most of the pathogenic variants were intronic (50%) and frameshift variants (50%). However, for *EPCAM* gene, both pathogenic variants were large deletions (100%) (Table 4).

On the other hand, one large deletion (deletion encompassing exons 6 to 11) was observed to be recurrent in 3 affected members in a homozygous state. The 3 members belong to the same tribe and were diagnosed with Constitutional Mismatch Repair Deficiency Syndrome (CMMRD) (Appendix A).

Four novel pathogenic variants have been reported in 7 affected individuals; two in *MLH1* gene (p.Thr545ProfsX46 and p.Thr553ProfsX38), one in *MSH2* gene (IVS11 + 2T > C) and one in *PMS2* gene (deletion encompassing exons 1 to 6) which was also found in unaffected individuals. (Appendix A).

### 3.3. Genotype-Phenotype Correlation

#### 3.3.1. Correlation with Mutated Gene

A Chi-square test was conducted to investigate the correlation of the mutated gene with the clinicopathological parameters. There was no statistically significant correlation between gender, ethnicity, age at diagnosis, tumor location, CRC side, grade of cancer, LV invasion, mucinous component, family history, histopathology, and tumor size with the type of mutated gene (Table 5).

#### 3.3.2. Correlation with Type Pathogenic Variants

A Chi-square test was conducted to investigate the correlation between the type of variant with the clinicopathological parameters A statistically significant correlation between the side of colon cancer and the type of variant has been found, however after Bonferroni correction, the association was not found to be statistically significant. Additionally no statistically significant association was found between any of the clinicopathological parameters and the type of variant (Table 6).

#### 3.3.3. Effect of Mutated MMR Genes on Protein Expression

Since IHC tests were performed for affected patients only, the effect of mutated MMR genes on the protein expression was analyzed in LS affected individuals only. Of the MMR pathogenic variants, 9 (45%) were due to *MLH1* germline variants, of these 9 found in our individuals, the majority (6/9) showed correspondence with the loss of the MLH1 protein and its heterodimer *PMS2*. For the remaining three cases, one showed a loss of expression of *PMS2* protein with a frameshift variant in *MLH1* gene classified as pathogenic (c.2252_2253delAA, p.Lys571SerfsX3), the second showed a loss of MLH1, PMS2, and MSH6 proteins, however, IHC data were not available for the third case. Regarding *MSH2*, also 9 (45%) pathogenic and likely pathogenic variants were detected. From these 9 variants, 7 resulted in the loss of MSH2 protein and its heterodimer *MSH6,* one resulted in the loss of MLH1 and MSH2 proteins, and one resulted in the loss of only MSH2 protein only. For *MSH6* gene, one patient had a pathogenic frameshift variant (c.3475delT, p.Tyr1159ThrfsX25) with intact MMR proteins. Finally, there was no data regarding the MMR proteins of patients with germline variants in *PMS2* gene (Table 7).

### 3.4. Assessment of Amsterdam/Bethesda Criteria

Data such as the family history, age at diagnosis, and tumor pathology were used to assess whether affected and unaffected individuals are meeting Amsterdam/Bethesda criteria. All affected patients (100%) strictly met either Bethesda or Amsterdam.

Regarding the eleven unaffected individuals, five (45.45%) were tested due to a family history of a known familial mutation, 3/11 (27.27%) were tested through panel genetic testing due to an extensive family history of multiple cancers, and 2 individuals (18.18%) were tested due to meeting Bethesda guidelines. However Only one individual (U007) had no family history of cancers nor a familial pathogenic variant but was found to be a carrier of a pathogenic variant in *PMS2* gene as a secondary finding of her whole-exome sequencing as part of the workup for hereditary glomerulosclerosis she has (Appendix A) (Table 8).

## 4. Discussion

### 4.1. Prevalence of Lynch Syndrome and Frequency of MMR Pathogenic Variants

To our knowledge, this is the first study to investigate the prevalence and genotype-phenotype correlation of Lynch syndrome in the state of Qatar. We investigated the prevalence of LS, frequency of pathogenic variants in MMR*/EPCAM* genes, and genotype-phenotype correlation in 31 individuals; 20 were affected with colon cancer and 11 were unaffected.

The prevalence of Lynch Syndrome among our selected CRC patients was found to be 22% (20/92). Compared to the studies from the MENA (Middle East and North Africa) region, this prevalence is less than what has been found in Pakistani selected patients (34.5%) as they have followed a very stringent criteria of selection (at least three relatives affected with LS associated cancers, at least one of them is a first degree relative of the other two, at least 2 different generations with LS associated cancers, individuals with cancer diagnosed at an age younger than 50 years) [22]. This prevalence of LS found in our patients is however, higher than most of the other reported CRC prevalence in the remaining MENA countries. For instance, in the Kingdom of Saudi Arabia (KSA) the prevalence of LS in CRC patients was 7% [23] and in Iran, it was 5.5% [24]. The prevalence of LS in our CRC patients was also higher compared to the prevalence reported by a study in the United States of America (USA) conducted mainly in the white population of Ohio (72/450) 16% [25].

The prevalence of Lynch syndrome in unaffected high-risk individuals was 2.2% (11/500), this prevalence was lower than that of unaffected high-risk relatives of patients with Lynch syndrome from Columbus, Ohio (USA) (77%; 102/132). Such high prevalence in the Ohio study is due to the fact that it was calculated among mainly first degree relatives of patients with confirmed pathogenic variants in one of the MMR genes and thus the possibility of detecting a pathogenic variant in a first degree relative is 50%. However, this is not the case for the high-risk unaffected individuals in the current study as not all of them had an affected relative confirmed to have pathogenic variants in MMR due to the fact that most didn’t pursue testing which could explain the lower prevalence of Lynch syndrome detected in our study compared to the previous Ohio study. Our approach to evaluate the prevalence of LS in selected high-risk patients/individuals affected or unaffected gives more insights regarding the genetic testing approach mainly for unaffected individuals. In this regard, our findings demonstrate the benefit of panel genetic testing for healthy high-risk individuals based on their family history especially when the types of cancers overlap in multiple genetic syndromes even though no pathogenic variant has been confirmed in relatives. Following, panel genetic testing approach, about 2.2% of unaffected relatives could be positive for any MMR gene variant and their identification could reduce their cancer risk through early surveillance, prophylactic prevention and early detection of cancers [26].

Among the affected individuals, the most reported genes were *MLH1* and *MSH2* accounting for 90% of the pathogenic variants, which is in agreement with earlier studies from Saudi Arabia [27] the United States [28], Finland [29], Spain [30] and the results of the international Mismatch Repair Consortium (IMRC) [15] This high prevalence of *MLH1* and *MSH2* genes in our cohort could also be attributed to the fact that affected patients were referred based on Amsterdam and Bethesda criteria, as families fulfilling these 2 criteria are more likely to harbor pathogenic variants in *MLH1* and *MSH2* genes [31] which might have resulted in missing some patients with pathogenic variants MSH6 and PMS2 genes. For unaffected LS individuals however, the most reported gene was *PMS2* accounting for 64% followed by *MSH6* and *EPCAM* genes with each accounting for 18% of the pathogenic variants. This high frequency of *PMS2* gene is due to the fact that all carriers were members of the same tribe. This finding suggests the presence of a potential tribal variant in *PMS2* gene which has a great clinical benefit as it can facilitate the selection of most of the at-risk individuals belonging to the same tribe for pre-test counseling and early cancer surveillance and pre-implantation genetic diagnosis. Such interventions would prevent further transmission of the pathogenic variant to the future generations. Additionally, the difference in the distribution of the genes between affected and unaffected individuals was statistically significant (*p* < 0.005) after Bonferroni correction, and the fact that *MLH1* and *MSH2* genes were not reported in unaffected individuals could be attributed to the high penetrance of these 2 genes and their association with higher cancer risk compared to the remaining MMR and *EPCAM* genes which makes them less likely to be detected in unaffected individuals [32].

With regards to the type of variants in each gene, for affected patients, point mutations and frameshift variants were commonly reported in *MLH1*, *MSH2*, and *MSH6* genes which goes in line with what has been reported in the Human Gene Mutation Database regarding the most common variant type in each gene (http://www.hgmd.cf.ac.uk/ac/all.php, accessed on 14 January 2022). For *PMS2* gene, however, one point mutation and one novel large deletion (deletion encompassing exons 6 to 11) were reported (Appendix A). A large deletion (deletion encompassing exons 6 to 11) was observed to be recurrent in 3 affected members of the same tribe in a homozygous state which is consistent with the diagnosis of Constitutional Mismatch Repair Deficiency Syndrome (CMMRD), a childhood-onset syndrome. Nevertheless, it is interesting to note the delay in onset of disease in these affected individuals (23, 25, and 28 years for patients C0021, C0022, and C0023 respectively) (Appendix A). This late-onset could be explained by the fact that CMMRD, *PMS2/MSH6* homozygous pathogenic variants are associated with a later onset phenotype compared to homozygous pathogenic variants in *MLH1/MSH2* genes which result in more aggressive hematological malignancies during young childhood and are associated with a worse prognosis [33]. A potential explanation for this late age at onset could also be the partial compensation of absent PMS2 by MLH3, which can form a functional heterodimeric protein with MLH1 that has mismatch repair capacity [34]. However, the surveillance for CMMRD is similar for all affected individuals regardless of which MMR gene was involved [35].

For unaffected individuals, the same previous recurrent large deletion (deletion encompassing exons 6 to 11) in *PMS2* gene has also been reported in a heterozygous state in 4 unaffected individuals belonging to the same tribe. Generally heterozygous pathogenic variants in *PMS2* display an attenuated Lynch syndrome phenotype consisting of lower penetrance and a later age at onset [36]. For *EPCAM* gene, all pathogenic variants were also large deletions which is in agreement with the nature of the common variants in this gene reported in other studies [6]. The heterozygous deletion of the entire *EPCAM* gene reported in patient U004 (Appendix A) and also previously reported in an individual with Lynch syndrome associated cancer [37], was reported of unknown significance on cancer risk because, it is known that deletions of 3′ region of *EPCAM* gene are associated with silencing of *MSH2* gene through the transcriptional read-through [38]. However, in the deletion we report in this study, the entire *EPCAM* gene is deleted and as a result transcription and transcriptional read-through might not occur, thus *MSH2* gene might not be affected which results in an unknown risk for Lynch Syndrome. However, it is considered pathogenic with respect to Congenital Tufting Enteropathy which is an autosomal recessive condition associated with biallelic pathogenic variants in the *EPCAM* gene [39]. Thereby, in the Qatari population where the consanguinity rate is high (54%) [40], the chances of serious autosomal recessive childhood-onset conditions such as CMMRD and Congenital Tufting enteropathy is increased especially with the presence of known Tribal variants. This suggests the necessity of implementing testing for targeted tribal variants in pre-marital screening and offering pre-implantation genetic testing for carrier parents to avoid the risk of autosomal recessive conditions associated with being homozygous for pathogenic variants in MMR/*EPCAM* genes [33,39].

### 4.2. Genotype-Phenotype Correlation in Affected LS Patients

There was no statistically significant association established between the type of cancer and mutated gene. One previous study found a lower expression of MLH1 protein in right sided colon cancer and a loss of MSH2 protein expression in the left sided colon cancers which might be due to germline mutations in the corresponding genes, however the association was not statistically significant [41].

Additionally, although it was expected that *MLH1* and *MSH2* genes would be associated with a younger age at onset as compared to *MSH6* and *PMS2* there was no statistically significant correlation (Table 5)

In line with what has been published in the literature [7,16], we did not find a statistically significant correlation between gender, ethnicity, age, tumor location, type of cancer, grade of cancer, lymph vascular, invasiveness, mucinous component, family history, histopathology, and tumor size with the type of mutated gene which might also be explained by our small sample size (Table 5).

It was expected that carriers of large deletions and frameshift variants would exhibit a more severe phenotype and an earlier age at onset compared to carriers of point mutations [42]. However, there was no statistical correlation between the type of variant and any of the clinicopathological parameters tested especially after Bonferroni correction for the association of CRC side with the type of mutated variant (Table 6). These findings are in agreement with the findings of a previous study on lynch syndrome patients from Spain where their correlation of the type of variant with clinicopathological variables did not yield any statistically significant results [7].

Immunohistochemistry testing was performed on tumors from all affected individuals with colon cancers as a first step (before the referral to the high-risk clinic) to test for the presence of MMR proteins and select individuals at high risk for LS. In our cohort, most of *MLH1* pathogenic variants, (6/9) (66.7%) had a corresponding loss of MlH1 protein expression along with the loss of its heterodimer PMS2. The loss of MLH1 and PMS2 proteins on IHC is expected as MLH1/PMS2 tend to form heterodimers. Regarding *MSH2*, (7 variants resulted in a corresponding loss of MSH2 protein expression and its heterodimer MSH6 protein, the loss of MSH2 and MSH6 proteins on IHC is due to the formation of heterodimers between these 2 proteins. However, the loss of MLH1 protein could be explained by the low sensitivity associated with some antibodies used in the IHC technique [11]. On the other hand, one pathogenic frameshift variant detected in *MSH6* gene resulted in intact MMR proteins expression. One of the limitations of IHC technique is that not all pathogenic MMR variants result in loss of immunoreactivity, for instance, frameshift and truncating variants can interfere with the protein function without altering the antigenic site of the protein [43], additionally, the interaction between MMR proteins could affect the sensitivity of immunohistochemistry in the detection of individual proteins [11].

Therefore, regardless of which protein was lost based on IHC, performing Panel testing is recommended for all individuals suspected to have Lynch syndrome [44] and not strictly meeting Bethesda or Amsterdam criteria. Clinicians should attentively take the personal and family history of the patient to be able to assess their eligibility for genetic testing. Evidently, the adoption of an effective screening program is challenging and is a topic of ongoing debate in the literature [27].

### 4.3. Assessment of Bethesda/Amsterdam Criteria

In the current study, 100% of the affected patients met either Bethesda 18/20 (91.3%) or Amsterdam criteria 2/20 (8.7%).

For unaffected individuals, only 2/11 (18.18%) met Bethesda guidelines, while 3/11 (27.27%) were tested using Panel genetic testing due to an extensive family history of cancers not meeting Amsterdam/Bethesda criteria (Table 8). These findings are suggestive of the high efficacy of Amsterdam/Bethesda criteria for affected high-risk individuals and not for unaffected high-risk individuals, thus Panel genetic testing is important for these unaffected individuals with family histories of multiple cancers even if they do not meet Amsterdam/Bethesda guidelines. Though the sensitivity and specificity of these two criteria could not be confirmed due to the lack of data regarding Lynch Syndrome negative patients. However, the fact that individuals meeting Bethesda guidelines were more than those meeting Amsterdam criteria is due to the flexibility of Bethesda compared to Amsterdam criteria [9].

However, a previous study by Syngal S. et al., (2000) [9] on 70 families with suspected hereditary colorectal cancer, found out that Amsterdam criteria for HNPCC were neither sufficiently sensitive nor specific for use as a sole criterion for determining which families should undergo testing. Additionally, they found out that the application of Bethesda guidelines was a less strict approach and was associated with a higher sensitivity [9].

### 4.4. Limitations

We believe that the small sample size of both affected and unaffected individuals considered in the present study could have had a significant effect on the statistical analysis and the generalizability of the study findings. Many Lynch syndrome patients could have been missed due to not being correctly identified and referred to genetic testing which would have increased the prevalence of LS. Furthermore, genetic testing for Lynch syndrome is currently not available in Qatar and many non-Qatari “eligible” patients could not afford it, in addition to the fact that some eligible patients who can afford testing but refuse to pursue it due to personal/social reasons especially unaffected patients who often express fear and anxiety of testing. Finally, the lack of clinicopathological data about the Lynch syndrome negative patients have limited a better assessment of the specificity/sensitivity of Amsterdam and Bethesda criteria

## 5. Conclusions

This study highlights the prevalence and genotype-phenotype correlation of LS in high-risk cohort of patients in the State of Qatar. The prevalence of LS among affected CRC patients and unaffected patients is 22% and 2.2% respectively which indicates an increased prevalence in the high-risk populations and draws recommendations on diagnostic and predictive genetic testing and personalize surveillance and management programs.

Among unaffected individuals, a recurrent *PMS2* pathogenic variant (deletion encompassing exons 6 to 11) was reported in several related individuals suggesting a tribal effect. This is indeed a significant finding which impacts our recommendations for at risk families from this high risk tribe, in which the rate of consanguineous marriage is high, who may benefit from preventative measures, risk-reducing strategies and premarital, prenatal and reproductive genetic counseling to reduce the risk of Lynch syndrome and serious autosomal recessive childhood-onset conditions such as CMMRD in these individuals.

Among affected individuals, *MLH1* and *MSH2* genes were highly frequent, thus drawing conclusions on the importance of establishing germline testing for all MMR’s especially *MLH1* and *MSH2* genes testing given its high prevalence and which will aide the existing somatic immunohistochemical staining (IHC) testing. Although IHC can aid germline testing, panel germline testing for all MMR’s is recommended to be available for all individuals suspected to have Lynch syndrome regardless of which protein is deficient and not strictly meeting Bethesda or Amsterdam criteria.

This is the first study of its kind in Qatar which serves as the foundation for more studies in the future concerning the epidemiology and genetics of Lynch syndrome in Qatar which can be captured at a bigger scale cohort of patients and at a population level.

## Figures and Tables

**Figure 1 genes-13-02176-f001:**
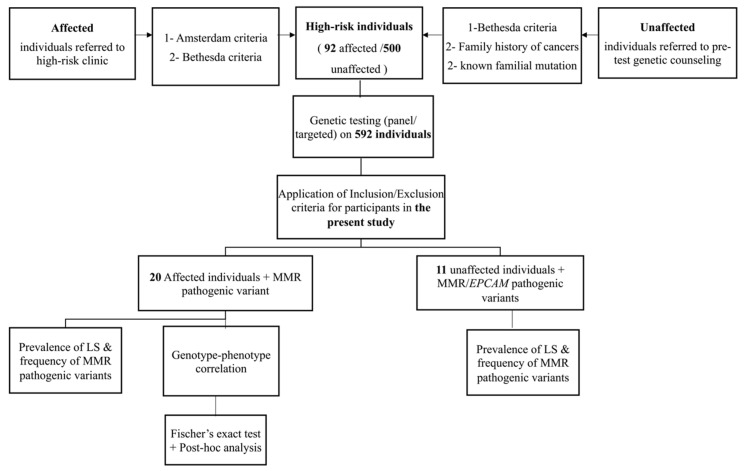
Flow chart of the study design. CRC: Colorectal Cancer. LS: Lynch Syndrome, MMR: Mismatch repair genes.

**Table 1 genes-13-02176-t001:** Affected LS patients’ demographics.

Patient Characteristics	N (%)
Age	
Mean	48.2
Range	(24–67)
Standard Deviation	±11.2
Age at diagnosis	
<50 y	14 (70%)
>50 y	6 (30%)
Total	20
Family History	
Yes	15 (75%)
No	5 (25%)
Total	20
Gender	
Male	14 (70%)
Female	6 (30%)
Total	20
Type of Cancer	
Right colon cancer	8 (40%)
Left colon cancer	12 (60%)
Total	20
Stage	
Stage 1	6 (30%)
Stage 2	12 (60%)
Stage 3	1 (5%)
Stage 4	1 (5%)
Total	20

**Table 2 genes-13-02176-t002:** Distribution of MMR genes variants in Affected and Unaffected individuals.

Mutated Gene						*p*-Value
	*MLH1*	*MSH2*	*MSH6*	*PMS2*	*EPCAM*	**<0.001 ^a^**
Unaffected individuals	0	0	2 (18.2%)	7 (63.6%)	2 (18.2%)	
Affected patients	9 (45%)	9 (45%)	1 (5%)	1 (5%)	0	
** *p* ** **-value ^b^**	0.009	0.009	0.230	0.000 ^c^	0.045	
Total	9	9	3	8	2	

^a^ = *p*-value before Bonferroni correction, using Fischer’s exact test. ^b^ = *p*-values after post hoc analysis. ^c^ = statistically significant *p*-values after Bonferroni correction (0.005).

**Table 3 genes-13-02176-t003:** Types of variants per MMR gene in Affected individuals.

Type of Variant	Mutated Gene			
	*MLH1*	*MSH2*	*MSH6*	*PMS2*
Frameshift mutations	4 (44.4%)	2 (22.2%)	1 (100%)	0
Point mutations	1 (11.1%)	5 (55.6%)	0	1 (100%)
Large deletions	2 (22.2%)	0	0	0
Intronic mutations	2 (22.2%)	2 (22.2%)	0	0
Total	9	9	1	1

**Table 4 genes-13-02176-t004:** Types of variants per MMR gene in unaffected individuals.

Type of Variant	Mutated Gene		
	*MSH6*	*PMS2*	*EPCAM*
Large deletions	0	5 (71.42%)	2 (100%)
Point mutations	0	1 (14.28%)	0
Intronic mutations	1 (50%)	1 (14.28%)	0
Frameshift mutations	1 (50%)	0	0
Total	2	7	2

**Table 5 genes-13-02176-t005:** Genotype- phenotype correlation with the mutated gene, LV: Lympho-vascular invasion, CRC: colorectal cancer.

	*MLH1*	*MSH2*	*MSH6*	*PMS2*	*p* Value
Gender					
Male	7 (77.8%)	7 (77.8%)	0	0	0.159
Female	2 (22.2%)	2 (22.2%)	1 (100%)	1 (100%)	
Total	9	9	1	1	
Ethnicity					
Qatari	0	0	1 (100%)	1 (100%)	0.641
Egyptian	2 (22.2%)	2 (22.2%)	0	0	
Sudanese	1 (11.1%)	1 (11.1%)	0	0	
Bangladeshi	2 (22.2%)	0	0	0	
British	1 (11.1%)	0	0	0	
Filipino	1 (11.1%)	1 (11.1%)	0	0	
Indian	1 (11.1%)	1 (11.1%)	0	0	
Nepalese	1 (11.1%)	1 (11.1%)	0	0	
Sri lankan	0	1 (11.1%)	0	0	
Pakistani	0	1 (11.1%)	0	0	
Palestinian	0	1 (11.1%)	0	0	
Total	9	9	1	1	
Age at diagnosis					
<50 y	8 (88.9%)	5 (55.6%)	0	1 (100%)	0.159
>50 y	1 (11.1%)	4 (44.4%)	1 (100%)	0	
Total	9	9	1	1	
Tumor Location					
Ascending Colon	3 (33.3%)	3 (33.3%)	0	0	0.883
Descending Colon	2 (22.2%)	0	0	0	
Rectum	1 (11.1%)	2 (22.2%)	0	0	
Sigmoid Colon	2 (22.2%)	2 (22.2%)	1 (100%)	1 (100%)	
Cecum	1 (11.1%)	1 (11.1%)	0	0	
Transverse colon	0	1 (11.1%)	0	0	
Total	9	9	1	1	
CRC side					
Right Colon Cancer	4 (44.4%)	4 (44.4%)	0	0	0.687
Left Colon Cancer	5 (55.6%)	5 (55.6%)	1 (100%)	1 (100%)	
Total	9	9	1	1	
Grade					
Stage 1	3 (33.3%)	2 (22.2%)	1 (100%)	0	0.783
Stage 2	5 (55.6%)	6 (66.7%)	0	1 (100%)	
Stage 3	0	1 (11.1%)	0	0	
Stage 4	1 (11.1%)	0	0	0	
Total	9	9	1	1	
LV invasion (yes/no)					
Yes	1 (14.3%)	0	0	0	0.712
No	6 (85.7%)	7 (100%)	1 (100%)	1 (100%)	
Total	7	7	1	1	
Mucinous component (yes/no)					
yes	2 (22.2%)	2 (50%)	1 (100%)	-	0.238
No	7 (77.8%)	2 (50%)	0	-	
Total	9	4	1	-	
Family History					
Yes	6 (66.7%)	7 (77.8%)	1 (100%)	1 (100%)	0.792
No	3 (33.3%)	2 (22.2%)	0	0	
Total	9	9	1	1	
Histopathology					
Well Differentiated adenocarcinoma	1 (14.3%)	2 (25%)	-	0	0.775
Moderately differentiated adenocarcinoma	6 (85.7%)	5 (62.5%)	-	1 (100%)	
Poorly Differentiated Adenocarcinoma	0	1 (12.5%)	-	0	
Total	7	8	-	1	
Tumor Size					
<5 cm	4 (57.1%)	4 (57.1%)	0	0	0.542
>5 cm	3 (42.9%)	3 (42.9%)	1 (100%)	0	
Total	7	7	1	0	

**Table 6 genes-13-02176-t006:** Genotype- phenotype correlation with the type of the variant, LV: Lympho-vascular invasion, Invasive: Invasive cancer based on pathology.

	Large Deletions	Intronic Mutations	Frameshifts	Point Mutations	*p*-Value
Gender					
Male	1 (50%)	4 (100%)	5 (71.4%)	4 (57.1%)	0.448
Female	1 (50%)	0	2 (28.6%)	3 (42.9%)	
total	2	4	7	7	
Age at diagnosis					
<50 y	2 (100%)	3 (75%)	5 (71.4%)	4 (57.1%)	0.691
>50 y	0	1 (25%)	2 (28.6%)	3 (42.9%)	
total	2	4	7	7	
Ethnicity					
Qatari	0	0	1 (14.3%)	1 (14.3%)	0.644
Egyptian	0	1 (25%)	3 (42.9%)	0	
Sudanese	1 (50%)	0	0	1 (14.3%)	
Bangladeshi	0	1 (25%)	1 (14.3%)	0	
British	0	0	1 (14.3%)	0	
Filipino	1 (50%)	0	0	1 (14.3%)	
Indian	0	1 (25%)	0	1 (14.3%)	
Nepalese	0	1 (25%)	1 (14.3%)	0	
Sri lankan	0	0	0	1 (14.3%)	
Pakistani	0	0	0	1 (14.3%)	
Palestinian	0	0	0	1 (14.3%)	
Total	2	4	7	7	
Tumor Location					
Ascending colon	0	4 (100%)	1 (14.3%)	1 (14.3%)	0.110
Descending colon	1 (50%)	0	1 (14.3%)	0	
Rectum	0	0	2 (28.6%)	1 (14.3%)	
Sigmoid colon	1 (50%)	0	3 (42.9%)	2 (28.6%)	
Cecum	0	0	0	2 (28.6%)	
Transverse colon	0	0	0	1 (14.3%)	
Total	2	4	7	7	
CRC Side					
Right Colon Cancer	0	4 (100%)	1 (14.3%)	3 (42.9%)	**0.026 ^a^**
Left Colon Cancer	2 (100%)	0	6 (85.7%)	4 (57.1%)	
*p*-value ^b^	0.23	0.01	0.09	0.84	
Total	2	4	7	7	
Grade					
Stage 1	0	1 (25%)	5 (71.4%)	0	0.088
Stage 2	2 (100%)	2 (50%)	2 (28.6%)	6 (85.7%)	
Stage 3	0	0	0	1 (14.3%)	
Stage 4	0	1 (25%)	0	0	
Total	2	4	7	7	
LV invasion (yes/no)					
Yes	0	0	1 (16.7%)	0	0.620
No	1 (100%)	3 (100%)	5 (83.3%)	6 (100%)	
Total	1	3	6	6	
Invasive (yes/no)					
Yes	1 (50%)	1 (33.3%)	3 (60%)	2 (50%)	0.912
No	1 (50%)	2 (66.7%)	2 (40%)	2 (50%)	
Total	2	3	5	4	
Mucinous component (yes/no)					
yes	0	0	2 (40%)	3 (75%)	0.138
No	2 (100%)	3 (100%)	3 (60%)	1 (25%)	
Total	2	3	5	4	
Family History					
Yes	2 (100%)	2 (50%)	6 (85.7%)	5 (71.4%)	0.480
No	0	2 (50%)	1 (14.3%)	2 (28.6%)	
Total	2	4	7	7	
Histopathology					
Well Differentiated adenocarcinoma	0	1 (33.3%)	1 (25%)	1 (14.3%)	0.883
Moderately differentiated adenocarcinoma	2 (100%)	2 (66.7%)	3 (75%)	5 (71.4%)	
Poorly Differentiated Adenocarcinoma	0	0	0	1 (14.3%)	
Total	2	3	4	7	
Tumor Size					
<5 cm	0	2 (66.7%)	3 (50%)	3 (60%)	0.688
>5 cm	1 (100%)	1 (33.3%)	3 (50%)	2 (40%)	
Total	1	3	6	5	

^a^ = *p*-value before Bonferroni correction, using Fischer’s exact test. ^b^ = *p*-values after post hoc analysis. Significant *p*-value after Bonferroni correction is <0.006.

**Table 7 genes-13-02176-t007:** MMR genes (*MLH1*, *MSH2*, and *MSH6*) with corresponding protein loss by IHC.

MMR Genes/Proteins	*MLH1*	*MSH2*	*MSH6*
MLH1-PMS2 proteins loss	6 (66.7%)	0	0
MLH1-PMS2-MSH6 proteins loss	1 (11.1%)	0	0
PMS2 protein loss	1 (11.1%)	0	0
MLH1-MSH2 proteins loss	0	1 (11.11%)	0
MSH2 protein loss	0	1 (11.11%)	0
MSH2-MSH6 Protein loss	0	7 (77.8%)	0
Intact MMR proteins	0	0	1 (100%)
Unknown status	1 (11.1%)	0	0
Total	9	9	1

**Table 8 genes-13-02176-t008:** Summary of criteria of selection of Affected and unaffected individuals.

Criteria	Affected	Unaffected
Amsterdam	2 (8.7%)	0
Bethesda	18 (91.3%)	2 (18.18%)
Familial mutation	0	5 (45.45%)
Through Panel testing	0	3 (27.27%)
As a secondary finding (through WES)	0	1 (9.09%)
Total	20	11

## Data Availability

Not applicable.

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
