# Peer review of "Prevalence and Genotype-Phenotype Correlation of Lynch Syndrome in a Selected High-Risk Cohort from Qatar’s Population"

_genes, 2022, doi:10.3390/genes13112176_

Round 1
Reviewer 1 Report
The purpose of this investigation was to determine the frequency, genotype-phenotype correlations, and survival rates associated with pathogenic variants of mismatch repair genes in Quatar. The study cohort of 35 individuals with pathogenic variants of DNA mismatch repair genes (24 with CRC and 11 unaffected) was ascertained from 592 individuals referred to a high-risk genetic clinic during January 2017 – August 2020. The findings add to the epidemiology of Lynch syndrome in the Quatari population. Some unique molecular features are evident, including the enrichment of a novel PMS2 deletion (exons 6 to 11), occurring in a homozygous and heterozygous state in affected and unaffected individuals, respectively.
Major comment:
One of the main findings regarding genotype-phenotype correlations was that MLH1 was associated with right-sided CRC and MSH2 with left-sided CRC (p=0.005). How did you define right-sided vs. left-sided? Table 5 shows the following distribution of bowel cancers for MLH1 pathogenic variant carriers: ascending colon 3, descending colon 2, rectum 1, sigmoid colon 2, cecum 1, duodenum 1, and transverse colon 0. This would make 4 right-sided CRCs (ascending colon, cecum) vs. 5 left-sided CRCs (descending colon, rectum, sigmoid colon). Duodenum is part of the small intestine and contributes to neither right-sided nor left-sided CRCs. Thus, the distribution seems identical to that of MSH2 pathogenic variant carriers. Please explain this discrepancy between the different pieces of data.
Minor comment:
Figure 2 shows Kaplan Meier survival analysis for the individual mismatch repair genes. The 5-year survival was 100% for MLH1, MSH2, and MSH6, whereas for PMS2 the survival was 70%. I assume PMS2 included both heterozygous and homozygous carriers of pathogenic variants? In the context of Lynch syndrome, only heterozygous variants should be included, since homozygosity for constitutional variants of mismatch repair genes gives rise to a different syndrome (CMMRD).
Author Response
Thank you for the valuable comments.
Please see the attachment.

Reviewer 2 Report
This paper is aiming to estimate the prevalence of Lynch syndrome in a selected, high-risk cohort of non-Caucasians in Qatar; to estimate the frequencies of pathogenic variants in the different mismatch repair genes; and to describe the genotype-phenotype correlation. It is a strength that data are national and collected over 3,5 years. Main contributions are documentation of frequencies of affected genes and types of pathogenic variants in non-Caucasian carriers of Lynch syndrome including a pathogenic variant causing both Lynch syndrome and CMMRD in a tribe with high degree of consanguineous marriages who could benefit from targeted genetic counselling and pre-reproductive testing.
General concept comments
The manuscript is generally well written, well structured and contributes data on an ethnic group with little data in the literature.
The very small sample size is, however, a major weakness. Any differences that may be between groups will most likely become statistically insignificant and the result will be little generalizable.
Multiple testing without correcting the calculation of p-values is a methodological grave mistake and must be corrected.
The cohort are biased by referral and the authors should address this in their interpretation and discussion of the results. It is stated that patients with colon cancer are referred if they fulfill the Amsterdam or Bethesda guidelines, the results show they do fulfill the guidelines, and the authors conclude that the guidelines are good at detecting Lynch syndrome. It is stated that unaffected individuals are referred if they have a positive family history of colon cancer or a known pathogenic variant in their family, the results show that only some fulfill the guidelines, and the authors conclude that the guidelines are not good at detecting Lynch syndrome in unaffected individuals.
The references are generally very old. Only 14 of 47 references have been published within the last 5 years, and I think the scientific value and relevance of the paper would be very much improved if the authors performed an up-to-date literature search and incorporated the newest evidence in the manuscript.
The Discussion could be improved by shortening and not repeating so many results
Specific comments
Introduction:
Line 42: I suggest referring to the revised Amsterdam and the revised Bethesda guidelines
Line 56-67: Should be updated. Could be with results from the International Mismatch Repair Consortium (IMRC) or the Prospective Lynch Syndrome Database (PLSD)
Study design would be easier to comprehend if some information were added and some information moved from “Study Participants” to “Study design”:
Line 89: Criteria for referral should be added
Line 91: How was the screening performed (by family history, IHC, germline panel testing, or?), and how was the testing for MMR genes performed (IHC, germline panel testing, or?)?
Line 101: pathogenic or likely pathogenic variants
Line 144: Bonferroni adjustment for multiple testing must be described here, performed, and the results presented in table 5 and 6, in the Results and Discussion
Results:
Line 167: “known pathogenic variant in a MMR gene in the family” instead of “known familial variant”
Table 2: More relevant to show frequencies in rows than in columns
Table 3 and 4: Type of variants in the first column should be identical in the two tables (Point variants/Point mutations etc.)
Line 198: “both pathogenic variants” is more appropriate than “all …”
Line 204-: Genotype-phenotype correlations will probably become insignificant after correcting for multiple testing and needing re-writing
207-8: “statistically significant difference between the location of colon cancer …” is confusing because p=0.911 for “Tumor location” in table 5. You mean significant between right and left side
Table 5: Totals are not correct for Gender (all genes) and Tumor Location (MSH2)
Table 6: Totals are not correct for Nationality
Table 5 and 6: P-values after correcting for multiple testing; maybe better heading then “Type of cancer” for right or left; LV invasion explained in footnote; “Invasive” to be defined; either Ethnicity or Nationality in both tables.
Table 7: “Intact MMR protein” instead of “Intact MMR proteins loss”
Line 256-: Survival analysis makes no sense in a cohort of 24 patients with 1 event
Discussion:
Could be improved by shortening and not repeating so many results
Line 272: MENA to be written in full first time for foreign readers
Line 288: Possibility of detecting a pathogenic variant is in fact less than 50%, which can be explained by Berkson’s bias
Line 302-317: High penetrance of MLH1 and MSH2 does also contribute to the observed high frequency among affected patients due to selection bias as families with pathogenic variants in MSH6 or PMS2 less often fulfill the Amsterdam or Bethesda guidelines
Line 329-337: Are affected children less likely to be referred than adolescents? If so, then that could also explain the observed late-onset of CMMRD
Line 372+388: check recalculated p-value
Line 379-: With such a small sample size, could you expect to find a statistically significant difference?
Line 397: Were all affected individuals with colon cancer IHC-tested before referral to your department (=universal testing) or after referral?
Line 399-: It would be much more interesting for the readers to get data on how many of the 592 referred individuals who had abnormal IHC-staining, and on how many fulfilled the Amsterdam and Bethesda guidelines. Please provide the data if you got it.
Line 425: Results due to referral bias (ascertainment bias) as stated in my general comments. Should be discussed.
Line 444: Denmark missing, Sweden mentioned twice
Author Response
Thank you for the valuable comments.
Please see attached the attachment.

Round 2
Reviewer 1 Report
In this revised version, the authors corrected their previous erroneous data on CRC location (Tables 5 and 6). As a result, there was no significant correlation between the affected MMR gene and clinicopathological variables (Table 5). Moreover, survival analysis was omitted which makes sense.
Major comment:
In genotype-phenotype correlation with the type of variant (Table 6), the authors report a significant correlation between the side of colon cancer and the type of variant, but the difference did not remain significant after Bonferroni correction. However, the classification of variant types needs a closer specification in the Materials and Methods section to be able to interpret the results correctly. It would be especially important to know which mutation subtypes the group “point mutations” includes. If it includes missense plus nonsense variants, the classification is not compatible with the anticipated functional consequence. The basic expectation is that missense alterations only substitute an amino acid, whereas nonsense changes result in an immediate stop of translation leading to truncated protein. Therefore, nonsense variants logically belong to the same group as frameshift variants, whereas missense changes should constitute a separate group.
Minor comment:
Please check the spelling of Bonferroni correction after Materials and Methods (it should be Bonferroni and not Benferroni).
Author Response
Thank you for your comments.
Please find attached.

Reviewer 2 Report
Only text editing needed: Several words and lines are repeated throughout the manuscript (e.i.: l. 497-517)
